# Imaging of glucose metabolism by 13C-MRI distinguishes pancreatic cancer subtypes in mice

Shun Kishimoto[1†], Jeffrey R Brender[1†*], Daniel R Crooks[2], Shingo Matsumoto[3,4], Tomohiro Seki[1], Nobu Oshima[1], Hellmut Merkle[5], Penghui Lin[6], Galen Reed[7], Albert P Chen[7], Jan Henrik Ardenkjaer-Larsen[7,8], Jeeva Munasinghe[5], Keita Saito[1], Kazutoshi Yamamoto[1], Peter L Choyke[9], James Mitchell[1], Andrew N Lane[6,10], Teresa WM Fan[6,10], W Marston Linehan[2], Murali C Krishna[1*]

[1]Radiation Biology Branch, Center for Cancer Research, NCI, NIH, Bethesda, United States; [2]Urologic Oncology Branch, Center for Cancer Research, NCI, NIH, Bethesda, United States; [3]Graduate School of Information Science and Technology, Division of Bioengineering and Bioinformatics, Hokkaido University, Sapporo, Japan; [4]JST, PREST, Saitama, Japan; [5]NINDS, NIH, Bethesda, United States; [6]Center for Environmental and Systems Biochemistry, University of Kentucky, Lexington, United States; [7]GE HealthCare, Chicago, United States; [8]Department of Electrical Engineering, Technical University of Denmark, Kongens Lyngby, Denmark; [9]Molecular Imaging Program, Center for Cancer Research, NCI, NIH, Bethesda, United States; [10]Markey Cancer Center, University of Kentucky, Lexington, United States

*For correspondence:
jeffery.brender@mail.nih.gov (JRB);
murali@helix.nih.gov (MCK)

[†]These authors contributed equally to this work

**Abstract** Metabolic differences among and within tumors can be an important determinant in cancer treatment outcome. However, methods for determining these differences non-invasively in vivo is lacking. Using pancreatic ductal adenocarcinoma as a model, we demonstrate that tumor xenografts with a similar genetic background can be distinguished by their differing rates of the metabolism of 13C labeled glucose tracers, which can be imaged without hyperpolarization by using newly developed techniques for noise suppression. Using this method, cancer subtypes that appeared to have similar metabolic profiles based on steady state metabolic measurement can be distinguished from each other. The metabolic maps from 13C-glucose imaging localized lactate production and overall glucose metabolism to different regions of some tumors. Such tumor heterogeneity would not be not detectable in FDG-PET.
DOI: https://doi.org/10.7554/eLife.46312.001

## Introduction

Altered metabolism to sustain rapid growth is one of the hallmarks of cancer (*Vander Heiden and DeBerardinis, 2017*). While certain common features of tumor metabolism are retained even in different cancers (*Ward and Thompson, 2012*; *Pavlova and Thompson, 2016*), other features can vary considerably among patients and even within a single tumor (*Hensley et al., 2016*; *Cros et al., 2018*). Since metabolite levels control a host of cellular processes from cell signaling to growth and differentiation (*Vander Heiden and DeBerardinis, 2017*; *Ward and Thompson, 2012*), local variations in metabolism can impact the sensitivity of the tumor to different forms of treatment. Local concentrations of glucose (*Liu et al., 2017a*; *Mathews et al., 2014*), fatty acids (*Svensson et al., 2016*), and amino acids (*Mathews et al., 2014*), for example, have been shown to influence the

efficacy of specific types of chemotherapy and radiotherapy, which could lead to a possible change of treatment strategy (*Daemen et al., 2015*). Personalizing treatment in response to fluctuations of metabolites requires a reliable way of measuring the local concentrations of metabolites, which is less well-established than techniques for measuring the cellular, genomic, and proteomic environment. The only molecular imaging technique in widespread clinical use, FDG-PET, is largely limited to measurements of glucose uptake only and cannot characterize downstream biochemical transformations (*Giordano et al., 2016*). This limitation is particularly important for organs like the brain which have a high natural glucose uptake. In these situations, measurements of downstream products like lactate become important in preventing false positives (*De Feyter et al., 2018*). Other techniques with high specificity, such as MALDI-MS, have been successful at characterizing tumor metabolic heterogeneity in biopsies (*Giordano et al., 2016*; *Mirnezami et al., 2014*) but cannot be employed in vivo. Tumor metabolism in vivo can be imaged by $^1$H magnetic resonance spectroscopy, but as a steady state method it contains contributions from multiple processes and cannot distinguish the internal metabolism of the tumor from contributions from the surrounding stroma.

Imaging exogenous metabolic tracers offers a more direct way of isolating biochemical pathways. In principle, this can be accomplished by MRI through the use of $^{13}$C labeled tracers. In practice, the signal from a $^{13}$C labeled tracer has been considered too weak for in vivo imaging. Hyperpolarized MRI was developed to surmount this obstacle, taking advantage of the fact that the high spin polarization of a paramagnetic radical can be transferred to the $^{13}$C nucleus on another molecule under resonant microwave irradiation, increasing the signal by three orders of magnitude or more. However, this transfer happens efficiently only at temperatures near ~1 K and the hyperpolarization is rapidly lost when the sample is brought to room temperature before injecting. Due to this limitation, these studies are normally applied to probes whose $^{13}$C T1 relaxation time is long enough that the enhanced polarization is not lost before the metabolic flux can be determined. Of these probes, pyruvate has proven to be one of the most useful probes as it interrogates the central switching point from glycolysis to the TCA cycle (*Gray et al., 2014*). By comparing the pyruvate-to-lactate conversion between tumors or between pre- and post-treatment, it has been possible to assess the glycolytic profile of tumors in vivo and assess metabolic flux changes during treatment (*Gutte et al., 2015*). However, hyperpolarized MRI using pyruvate is unable to detect changes occurring upstream of the TCA cycle, which are common in many cancers.

An alternative approach that allows a more comprehensive analysis is to use glucose as a metabolic tracer (*De Feyter et al., 2018*; *Rodrigues et al., 2014*; *Timm et al., 2015*). Although glucose itself is difficult to hyperpolarize, new techniques allow the dynamic imaging of metabolic tracers by MRI without hyperpolarization. This imaging clearly suffers from low signal sensitivity but this can be compensated for by efficient noise suppression (*Brender et al., 2019*). To see how a targeted approach using hyperpolarized pyruvate compares to the more comprehensive approach offered by non-hyperpolarized glucose, we analyzed the tumor xenografts of two closely related cell lines with a similar genetic background in metabolic properties. MIA Paca-2 and Hs 766T are cell lines established from pancreatic ductal adenocarcinomas (PDACs) with similar mutations in major metabolic genes (*Deer et al., 2010*). However, other differences in the anatomy of the xenografts and in non-metabolic properties of the cell lines can impact the tumor microenvironment. Hs 766T was derived from a metastatic site and is expected to have a different stromal boundary compared to MIA Paca-2, which is derived from a primary heterogeneous tumor. Hs 766T tumor xenografts are more hypoxic than MIA Paca-2 (*Zhang et al., 2016*), have a more poorly developed vasculature, and was expected to have a different overall physicochemical environment as a result of this anatomical difference. While MIA Paca-2 and Hs 766T have similar overall metabolism (*Daemen et al., 2015*), it is possible to detect a difference in glucose metabolism using a newly developed technique to image glycolysis using non-hyperpolarized $^{13}$C glucose as a tracer (*Brender et al., 2019*). Imaging local metabolite concentrations and biochemistry in this manner may provide a new method for understanding the tumor biochemical microenvironment.

## Results

### MIA Paca-2 and Hs 766T PDAC xenografts have distinct anatomical and histological characteristics

*Figure 1A and C* show transverse slices from the anatomical T2-weighted RARE MRI of xenografts of Hs 766T and MIA Paca-2. Both tumors are poorly differentiated and show the gross anatomy typical of Grade 3 PDACs (*Matsumoto et al., 2018*). While the gross anatomy is similar, the anatomical microstructure and histology of the two tumors is distinctly different. The MIA Paca-2 tumors appear entirely homogenous and undifferentiated, an observation that holds down to the cellular level (*Figure 1.D*). By contrast, the homogeneity of the Hs 766T tumors is broken by hypointense spots, a feature characteristic of focal necrosis (*Figure 1.B*) (*Jakobsen et al., 1995*). As noted in previous reports (*Bailey et al., 2014*), we found similar levels of CD31, a common biomarker for angiogenesis (*Figure 1—figure supplement 1*), (*DeLisser et al., 1997*) suggesting immature, rather than deficient, vasculature may be responsible for the higher hypoxia levels in Hs 766T (*Bailey et al., 2014*). At the cellular level, cell rupture and inflammation were evident in Hs 766T but not in MIA Paca-2 cells (*Figure 1B and D* arrows). Despite their overall genetic similarity, the tumor microenvironment differs and Hs 766T and MIA Paca-2 can be easily distinguished by either anatomical MRI or histology.

### PDAC xenografts can easily be differentiated from non-cancerous host tissue by metabolic differences

Both the MRI and histology results point to substantial differences in the tumor microenvironment between the two tumor types that may influence metabolism. Accordingly, we looked for alterations in central metabolic pathways for biosynthesis, stress response, and energetics that are commonly modified in tumor cell lines using capillary electrophoresis mass spectrometry (CE-MS) targeted metabolic profiling (*Matsumoto et al., 2014*).

As expected, pancreatic host tissue from the mouse was metabolically distinct from Hs 766T and MiaPaca2 xenografts ($p < 0.001$ based on two-way ANOVA), with numerous metabolic differences across multiple metabolic pathways (*Figure 1—figure supplement 2*). The largest changes are concentrated in pathways connected to amino acid biosynthesis and degradation, reflecting an imbalance between amino acid metabolism and protein biosynthesis caused by unsustainable growth. Amino acid levels of all types and amino acid synthetic intermediates were strongly attenuated in both cell lines. Both cell lines also show a strong depletion in the level of intermediates throughout the urea cycle, the primary pathway for protein catabolism, as well as the polyamine biosynthetic pathway downstream of the urea cycle.

Major metabolic changes are also evident in other pathways. Consistent with previous reports on PDAC tumors (*Ying et al., 2012*), steady state concentrations of glycolytic intermediates were elevated in both cell lines, so were the metabolites of the pentose phosphate pathway for nucleotide and NADPH production (*Figure 1—figure supplement 3*). To counter the increased oxidative stress, there were increased levels of metabolites in the methionine redox cycle of the tumor xenografts: the reduced equivalents are depressed and the oxidized equivalents elevated compared to normal tissue. Finally, lactate levels are also highly elevated (*Wang et al., 2014*), consistent with a Warburg phenotype for both Hs 766T and MIA Paca-2 tumors.

### Few metabolic biomarkers distinguish MIA Paca-2 and Hs 766T

The metabolic differences between Hs 766T and MiaPaca2 PDAC tumors were more subtle. Although it is possible to distinguish between the two types of PDAC tumors using the entirety of the metabolic profile ($p = 0.00015$ for N = 4, two-way ANOVA with Sidak's correction for multiple comparisons), no single pathway stood out as being distinct (*Figure 1E*). Only a few biomarkers are distinct at the 5% confidence level with most of the differences that do exist are in the TCA cycle. The most striking difference was in fumarate levels ($p = 0.003$), which were significantly depressed (decreased fourfold relative to normal) in MIA Paca-2 and normal or slightly elevated in Hs 766T (elevated 1.4-fold). Fumarate has been suggested as an oncometabolite (*Yang et al., 2012*; *Sullivan et al., 2013*) created both through the TCA cycle and as a byproduct of the urea cycle that competitively inhibits 2-OG-dependent oxygenases to stabilize the HIF complex and induce

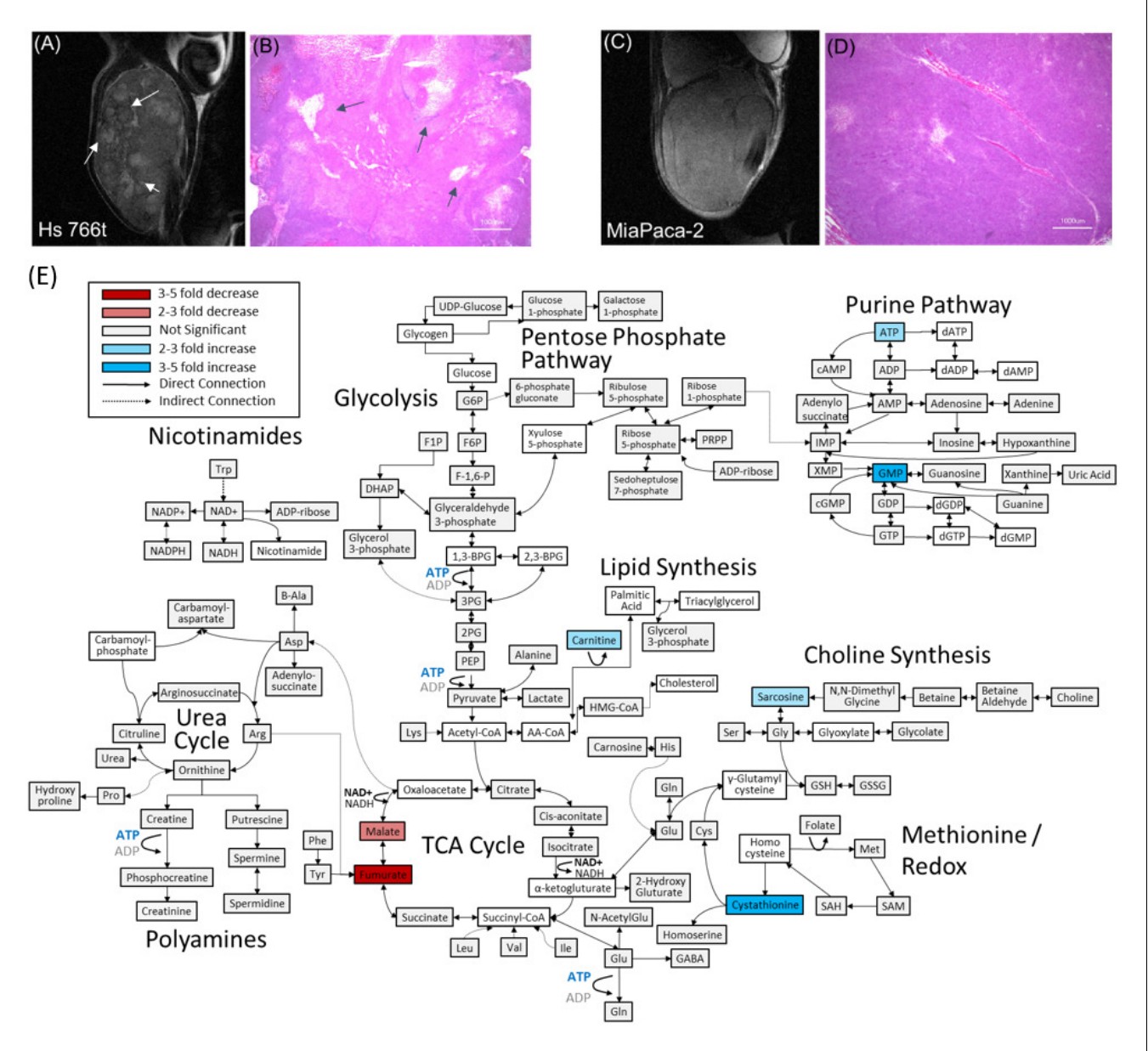

**Figure 1.** Distinct anatomical microstructure differences exist between Hs 766T and MIA Paca-2 xenografts while CE-MS shows only subtle metabolic differences. (A and C) T2 weighted anatomical RARE images of (A) Hs 766T and (C) MIA Paca-2 PDAC xenografts implanted on the left leg. Focal necrosis is evident in the Hs 766T tumor, but not in the MIA Paca-2 one. (B and D) H and E staining of biopsies from (B) Hs 766T and (D) MIA Paca-2 tumors. Cell rupture is present in the Hs 766T biopsy. (E) Metabolite differences of MIA Paca-2 and Hs 766T PDAC leg xenografts as analyzed by CE/MS. White boxes indicate metabolites not detected. Gray boxes indicate a statistically insignificant difference between cell lines (two-sided t-test, corrected for multiple comparisons by the two-stage linear step-up procedure of Holm et al with a confidence level of 5%). Blue and red boxes indicate statistically significant increases or decreases with respect to MIA Paca-2.

DOI: https://doi.org/10.7554/eLife.46312.002

The following figure supplements are available for figure 1:

**Figure supplement 1.** Protein expression levels from immunoblotting of tumor extracts of key proteins associated with metabolism.
DOI: https://doi.org/10.7554/eLife.46312.003

**Figure supplement 2.** Metabolite differences between normal tissue and the MIA Paca-2 PDAC leg xenografts as analyzed by CE/MS.
DOI: https://doi.org/10.7554/eLife.46312.004

**Figure supplement 3.** Metabolic differences within glycolysis between normal tissue and MIA Paca-2 and Hs 766T PDAC leg xenografts in terms of absolute concentrations (nmol/g tumor wet weight).
DOI: https://doi.org/10.7554/eLife.46312.005

pseudohypoxia. Malate and arginosuccinate, two other intermediates linked to fumarate metabolism were also significantly depressed in MIA Paca-2.

## Pyruvate metabolism is indistinguishable between PDAC hypoxic subtypes

The CE/MS experiment measures the static distribution of metabolites within the tumor, which is the sum of multiple biochemical pathways. While the data suggests that a difference in glycolysis and oxidative phosphorylation may exist between the MIA Paca-2 and Hs766 cell lines, the statistical significance of these changes is mostly uncertain and the origin of the effect is not clear - it is uncertain whether the difference is the result of upregulation of specific genes or is a more general effect from changes in the underlying physiology of the tumor microenvironment. To more directly probe specific enzyme activities within the glycolytic and TCA cycles, we tracked the in vivo utilization of hyperpolarized $^{13}$C labeled pyruvate using magnetic resonance spectroscopy to detect the de novo generation of new metabolites from pyruvate. Pyruvate metabolism is a central control point between glycolysis and oxidative phosphorylation and dysregulation of pyruvate dehydrogenase can be an important component of the Warburg effect (*Saunier et al., 2016*).

*Figure 2A and B* shows typical spectra after the injection of 98 mM solution of hyperpolarized [1-$^{13}$C] pyruvate into the tail vein of nude mice bearing MIA Paca-2 or Hs 766T xenografts in the left leg. The five observed peaks correspond to pyruvate (172.6 ppm), lactate (184.9), alanine (178.2), bicarbonate (162.6 ppm), and inactive pyruvate hydrate (180.9 ppm). Few differences could be seen when using C-1 labeled pyruvate as a metabolic tracer; pyruvate metabolism appears to be statistically indistinguishable in the MIA Paca-2 and Hs 766 T cell lines. The rate of pyruvate to lactate conversion was similar in MIA Paca-2 and Hs 766T as was both the rate of transamination of pyruvate to alanine and the first step of the TCA cycle as measured by bicarbonate production (see *Figure 2*). While differences in pyruvate metabolism in hypoxic and oxidative tumors have previously been shown by hyperpolarized C-1 labeled pyruvate (*Matsumoto et al., 2018*), pyruvate metabolism is not a sensitive biomarker for distinguishing subtle differences among hypoxic pancreatic adenocarcinoma subtypes.

## Glucose metabolism can be measured in vivo by $^{13}$C MRS

The CE/MS data is suggestive of an upregulation in MIA Paca-2 of the later stages of glycolysis relative to Hs 766T (*Figure 1—figure supplement 3*), although no single metabolite in the glycolytic pathway stands out as being statistically different (*Figure 1*). The CE/MS assay, however, requires a biopsy, which may be undesirable in some circumstances. To probe glucose metabolism in vivo, a different technique is needed.

Hyperpolarized glucose imaging has been successfully used to image glycolysis in vivo (*Rodrigues et al., 2014*). While glucose is an exemplary tracer from a biological standpoint, its short relaxation time can be problematic for hyperpolarization (*Harada et al., 2010*), and the hyperpolarization process places constraints that may be difficult to realize in some situations in a clinical setting. Hyperpolarization of $^{13}$C tracers has been considered necessary for imaging and kinetic studies due to the inherently low signal of $^{13}$C MRS. If noise can be reduced to acceptable levels without hyperpolarization, the restrictions hyperpolarization places on an experiment can be removed. We have previously shown that it is possible to use the correlation of the $^{13}$C signal in both time and space to reduce the noise level in the signal by an order of magnitude or more without sacrificing accuracy (*Brender et al., 2019*). Noise reduction by rank reduction makes it possible to do away with the hyperpolarization step and the limitations it puts on the metabolic probes that can be used.

Using this technique, we first checked the glucose metabolism of each tumor type following an injection of 50 mg bolus of [U-$^{13}$C] glucose using non-localized spectroscopy. The resulting spectra are complex and include contributions from the α and β anomers of glucose, the natural abundance $^{13}$C signal from lipids, as well as signals from downstream metabolites such as lactate and alanine (*Figure 3A*). To help resolve these ambiguities, tumors were flash frozen 1 hr after the injection of a [U-$^{13}$C] glucose bolus and the polar fraction analyzed by multidimensional NMR. Strong signals at 22.8 and 18.9 ppm in the indirect dimension of the HSQC spectrum (*Figure 3—figure supplement 1A*) and carbon satellites in the TOCSY spectrum (*Lane and Fan, 2007*) (*Figure 3—figure supplement 1B*) confirmed that the peaks at 22.8 ppm and 18.9 ppm arise from $^{13}$C labeled lactate and

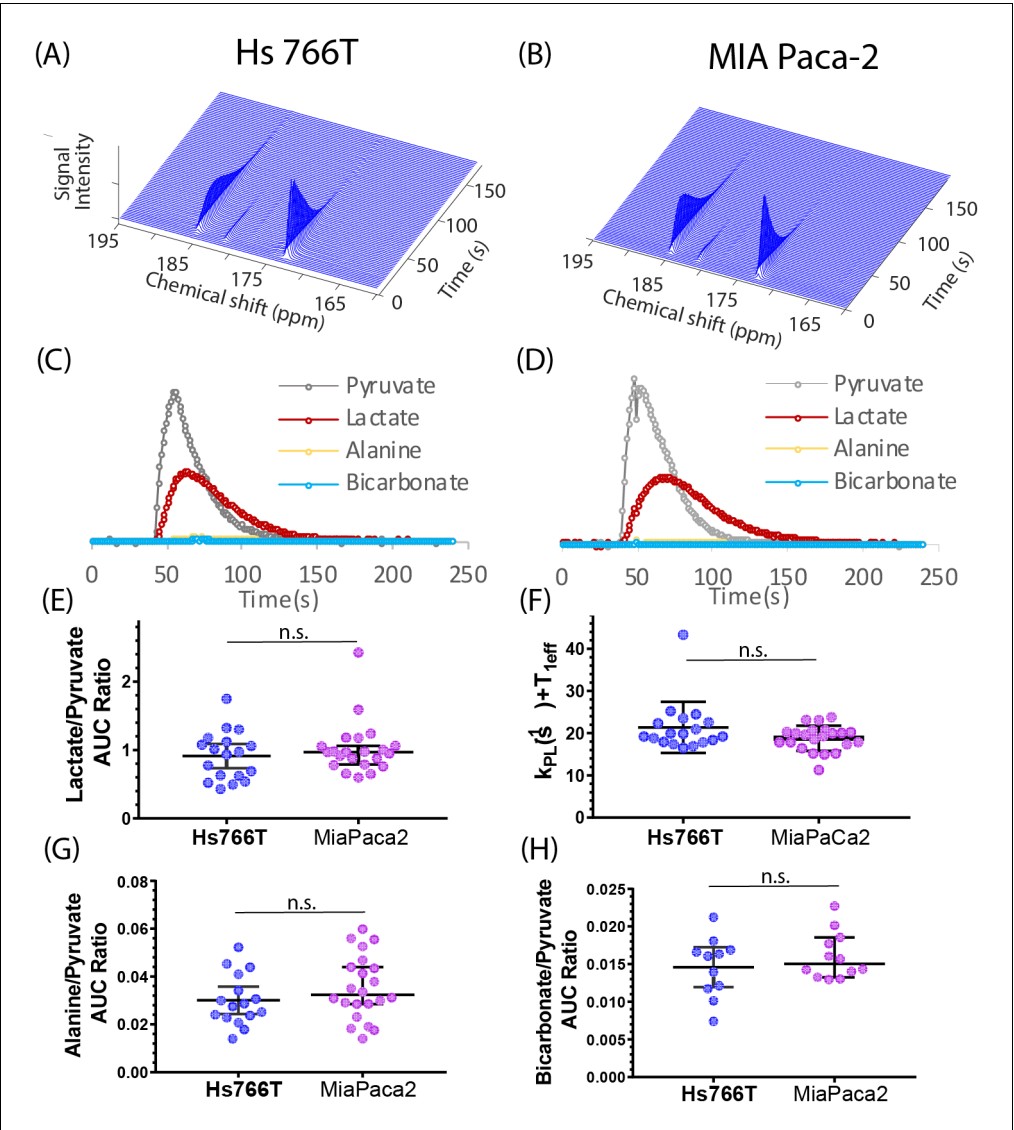

**Figure 2.** Pyruvate metabolism is similar in Hs 766T and MIA Paca-2 Xenografts. (**A and B**) Representative signal after injecting 300 µL of 98 mM hyperpolarized [1-$^{13}$C] pyruvate into the tail vein of a mouse of a nude mouse with either a (**A**) Hs 766T or (**B**) MiaPaCa2 leg xenograft. Signal loss is due to a combination of the loss of hyperpolarization and conversion of pyruvate to other metabolites. Corresponding kinetic traces of the pyruvate, lactate, alanine and bicarbonate signals metabolites for (**C**) Hs 766T and (**D**) MiaPaCa2 xenografts. (**E**) Ratio of the integrated lactate and pyruvate for Hs 766T (n = 18) and MiaPaca2 (n = 22) mice. The ratio is equal to the net lactate to pyruvate conversion rate in the absence of lactate efflux or back conversion. (**F**) Decay rate of the pyruvate signal, equivalent to the sum of the net lactate to pyruvate conversion rate and the effective relaxation rate (T$_{1eff}$, assumed to be the same between cell lines). (**G and H**) Ratio of the integrated alanine (**G**) or bicarbonate (**H**) to pyruvate. No statistically significant difference between cell lines was detected for any measure (Mann-Whitney rank test). Error bars represent 95% confidence intervals.

DOI: https://doi.org/10.7554/eLife.46312.006

alanine, respectively (*Figure 3A*). $^{13}$C labeled glutamate and aspartate are detectable in the ex vivo HSQC but not in the in vivo MRS, likely because the longer relaxation times (*Badar-Goffer et al., 1990*; *Allouche-Arnon et al., 2013*) strongly attenuate the signal with the short recycle delay used in the in vivo experiment. The peaks at 95 and 98 ppm can potentially arise from carbon 1 of the α and β anomers of either glucose or glucose-6-phosphate. The absence of any detectable peaks near 75 ppm confirms they arise exclusively from glucose without contribution from glucose-6-phosphate

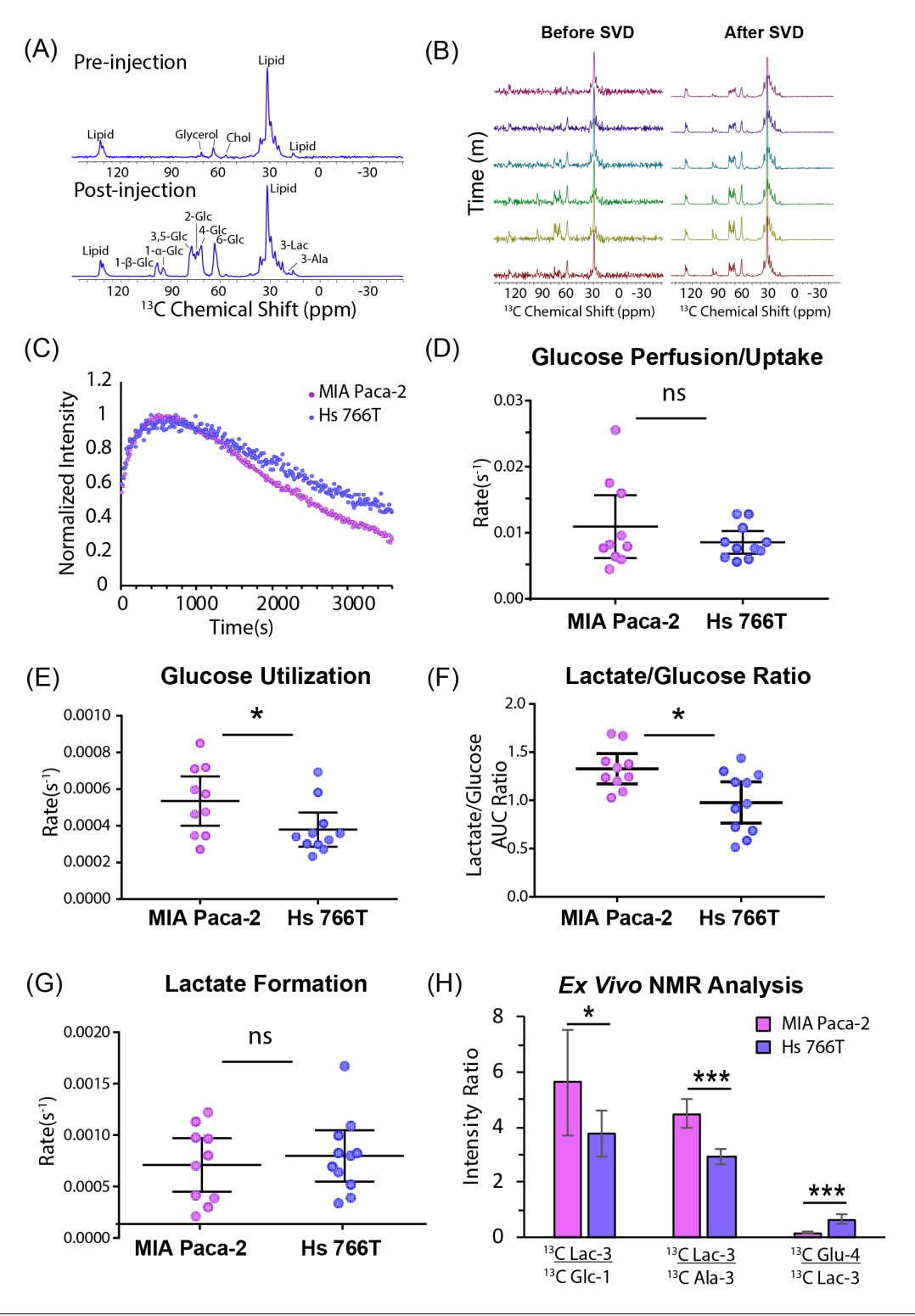

**Figure 3.** Glucose metabolism differentiates Hs 766T and MiaPaca2 xenografts. (**A**) Representative signal before and 1 hr after injecting 50 mg of [U-$^{13}$C] glucose into the tail vein of a mouse of a nude mouse with a MIA Paca-2 leg xenograft. The largest signal at 32 ppm is due to endogenous lipids, while glucose peaks can be seen in region from 60 to 100 ppm and lactate and alanine peaks can be detected at 22.8 ppm and 18.9 ppm, respectively. (**B**) Representative signals before and after rank reduction by SVD to reduce noise. No hyperpolarization is used in this experiment; the time dependence of the signal is due to metabolic

*Figure 3 continued on next page*

*Figure 3 continued*

interconversion. Spectra were acquired every 16 s (16 averages per scan). (C) Representative kinetic traces for the glucose signal at 98 ppm for Hs 766T (n = 10) and MIA Paca-2 (n = 11) xenografts. (D–F) Whether expressed as directly as rate obtained from curve-fitting (E) or as a ratio (F), a statistically significant difference in the rate of glucose metabolism can be seen between the Hs 766T or MIA Paca-2 PANC subtypes (Mann Whitney rank test, p=0.05 and 0.03 respectively). No statistically significant difference could be seen in the rate of glucose uptake (D) or lactate production (G). (H) Ex vivo NMR analysis of glucose and glucose-derived metabolites in the polar extract of the tumor xenografts (n = 5 for each group). Error bars in all cases represent 95% confidence intervals.

DOI: https://doi.org/10.7554/eLife.46312.007

The following figure supplement is available for figure 3:

**Figure supplement 1.** Ex vivo NMR analysis of Hs766T and MIA Paca-2 xenografts.

DOI: https://doi.org/10.7554/eLife.46312.008

---

or any other glycolytic intermediates (*Timm et al., 2015*; *Wishart et al., 2018*; *Wishart et al., 2007*). While the peaks at 95 and 98 ppm can be definitively assigned to carbon 1 of glucose, the HSQC and the pre-injection spectra (*Figure 3A*) confirms the other intense peak at 63 ppm and the spectrally crowded region between 63 and 78 ppm contains contributions from glycerol containing species. The lack of $^{13}C$-$^{13}C$ couplings for the glycerol resonances in the HSQC show that these molecules were not derived directly from the glucose bolus.

The peak at 98 ppm was therefore used as a marker as it can be assigned specifically in this case to glucose and not any other molecule. Specifically, an increase in the intensity of the 98 ppm resonance reflects the arrival of glucose from the bloodstream into the instrument's field of view (gluconeogenesis within the tumor is assumed to be negligible on this time scale). A decrease in intensity reflects either the removal of glucose by the bloodstream out of the field of view or the conversion of glucose into other species. Similarly, the appearance of peaks at 22.8 and 18.9 ppm reflect the conversion of lactate and alanine, respectively, and/or the arrival of circulating lactate or alanine produced from $^{13}C$ glucose outside the tumor (*Hui et al., 2017*; *Faubert et al., 2017*).

## Glucose metabolism, but not glucose uptake, distinguishes PDAC hypoxic subtypes

The improvement in temporal resolution afforded by the greatly increased signal-to-noise ratio allows an assessment of the fast glucose import step by MRI. The kinetics of glucose import have not been resolved effectively by $^{13}C$ MRI previously and are difficult to measure even with PET imaging (*Kuntner, 2014*). No statistically significant difference between cell lines could be detected in the rate of glucose uptake (*Figure 3D*), in agreement with the similar levels of the glucose transporter GLUT1, detected by western blot (see *Figure 1—figure supplement 1*), or in the rate of lactate formation (*Figure 3G*). The rate of glucose metabolism after import; on the other hand, distinguishes MIA Paca-2 and Hs 766T xenografts. Hs 766T xenografts displayed a statistically significant slower glucose metabolism than MIA Paca-2 xenografts (*Figure 3E*, Mann-Whitney rank test, p=0.02, Cohen's d = 3.43, large effect size). This difference is also reflected in the time-averaged glucose to lactate ratio (*Figure 3F*, Mann-Whitney rank test, p=0.03, Cohen's d = 1.20, large effect size), which is an approximate measure of the relative rates of the appearance of lactate and disappearance of glucose within the FOV of the scanner (*Hill et al., 2013*). To confirm the presence of a metabolic difference, the glucose, lactate, and glutamate peaks of the HSQC spectra from the polar extracts of MIA Paca-2 and Hs 766T tumors were quantified (*Figure 3H*). As expected, the metabolite ratios from ex vivo NMR do not exactly match those observed in the in vivo MRS technique because of the strong bias toward fast relaxing species such as glucose in the MRS spectrum due to the short repetition time. Nevertheless, the predicted trend (higher lactate to glucose ratios in MIA Paca-2) is similar in both experiments.

## Glucose imaging by $^{13}C$ MRS detects local differences in metabolism within PDAC tumors

*Figure 1E* shows that a metabolomic profile from a CE/MS can distinguish between xenografts of the MIA Paca-2 and Hs 766 T cell lines. Unfortunately, mass spectrometry can only be done ex situ and it is impossible to apply in a living tumor. The MRI spectroscopy experiment in *Figure 3* can

efficiently differentiate between MIA Paca-2 and Hs 766T xenografts but does not give information on local variations within the tumor. The low rank reconstruction procedure used in *Figure 3* can be extended into higher dimensions for imaging by tensor decomposition to give 30-fold or higher increases in signal to noise in spectroscopic imaging. As a first test, we used chemical shift imaging to provide a low-resolution (3 × 3×16 mm) map of the rates of glycolysis and anaerobic fermentation (*Figure 4A–D*). Simple chemical shift imaging was used to minimize potential imaging artifacts; considerable acceleration can be achieved by a more efficient pulse sequences and is the focus of ongoing research.

Figure 4 shows representative results from chemical shift imaging of MIA Paca-2 and Hs 766T xenografts before and after noise suppression (see Materials and methods). While the raw images are almost entirely noise, (*Figure 4A and E*) the processed images by tensor decomposition clearly show localized uptake of glucose within the tumor and conversion to lactate. As in the non-localized

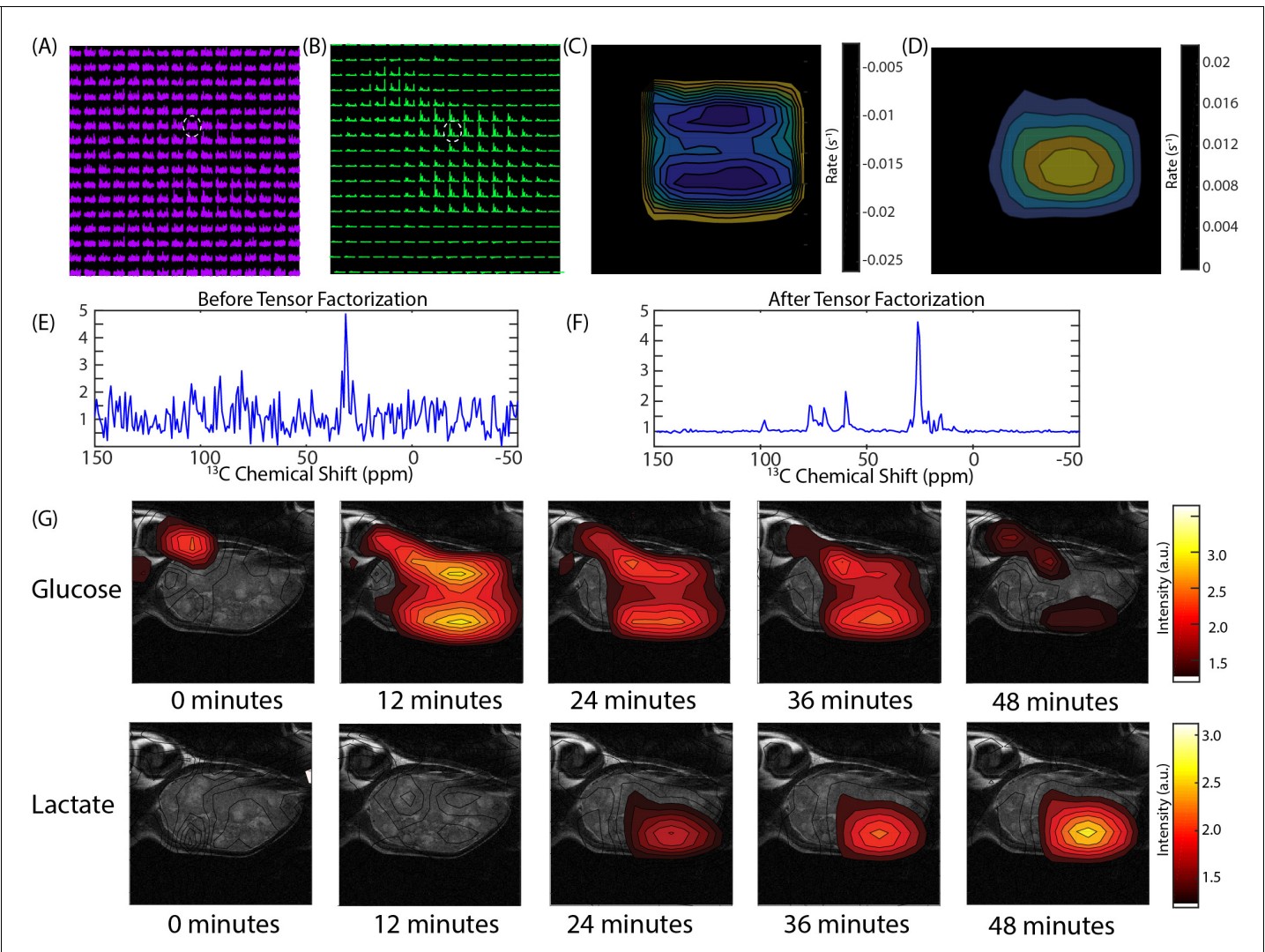

**Figure 4.** CSI imaging of a Hs 766T mouse leg xenograft after a 50 mg [U-13C] glucose injection in a volume of 300 microliters of PBS. An 8 × 8 image of the tumor bearing mouse leg was acquired by chemical shift imaging every 48 s for 60 min. The final image was zero-filled to 16 × 16. Each is voxel 0.15 cm x 0.15 cm x 1.6 cm in size. (**A**) The glucose region of the spectra at 12 min overlaid on the anatomical image. (**B**) Same image after tensor factorization. (**C and D**) Rate map of (**C**) glucose and (**D**) lactate metabolism calculated from the image series (**E and F**). Signal from the voxel indicated by the white dashed line (**E**) before and (**F**) after tensor factorization. (**G**) Contour maps created from the peak maximums of the glucose and lactate signals at the time points indicated.

DOI: https://doi.org/10.7554/eLife.46312.009

experiment, the glucose signal can be seen to decay and the corresponding lactate signal at 23 ppm to simultaneously increase as the tumor metabolized the bolus (*Figure 4G*). Local differences in metabolism can be detected in many tumors. For example, in one Hs 766T xenograft (*Figure 4G*) glucose metabolism is distributed relatively uniformly after taking into account the overall tumor anatomy. Lactate production, on the other hand, is localized in this tumor to one side where focal necrosis is more evident. In comparable MIA Paca-2 tumors (*Figure 5*), glucose uptake and lactate production appear to be more tightly correlated, congruent with the greater homogeneity apparent in the anatomical MRIs.

## Discussion

Pancreatic ductal adenocarcinoma (PDAC) represent 90% of pancreatic cancers and are character-ized by a poor prognosis and limited treatment strategies (*Hidalgo, 2010*; *Garrido-Laguna and Hidalgo, 2015*). Given PDACs resistance to traditional chemo- and radiotherapy regimes (*Adamska et al., 2017*), alternative points of attack are being considered. One potential point of attack is the dysregulated metabolism of PDACs (*Blum and Kloog, 2014*), which is highly dependent on protein autophagy and catabolism (*Yang et al., 2011*; *Kamphorst et al., 2015*) and exogenous glutamine and glucose (*Ying et al., 2012*; *Biancur et al., 2017*). Further, PDAC tumors usually have alterations in the activity in the urea cycle to support pyrimidine and amino acid synthesis (*Rabinovich et al., 2015*; *Liu et al., 2017b*) and often display a Warburg phenotype of increased glycolysis followed by diversion to lactate (*Anderson et al., 2017*). Each alteration and dependency represents a potential point of intervention. Although targeting the master genetic switches for these transformations, p53 and kRAS, (*Ying et al., 2012*; *Bryant et al., 2014*) is difficult, the down-stream enzymes are practical targets. Inhibitors for lactate dehydrogenase (*Rai et al., 2017*) and the lactate transporter MCT1 (*Sonveaux et al., 2008*) have shown promise in preclinical trials and may enter clinical trials in the near future. Beyond the Warburg effect, researchers have begun to target other vulnerable aspects of PDAC metabolism such as amino acid synthesis (*Ananieva, 2015*) or the unique chemical environment of tumors by hypoxia activated prodrugs (*Bailey et al., 2014*; *Liu et al., 2012*).

Targeting aberrant metabolism requires a method of monitoring treatment progress and select-ing suitable patient populations. Response to cancer treatment can be highly variable and there is a concerted push to tailor treatment regimens to individual patients (*Jain, 2005*). For protein targets such as receptors, genome sequencing or protein expression profiling is often sufficient to demon-strate a patient has a vulnerable mutation. Targeting aberrant metabolism is more difficult as the

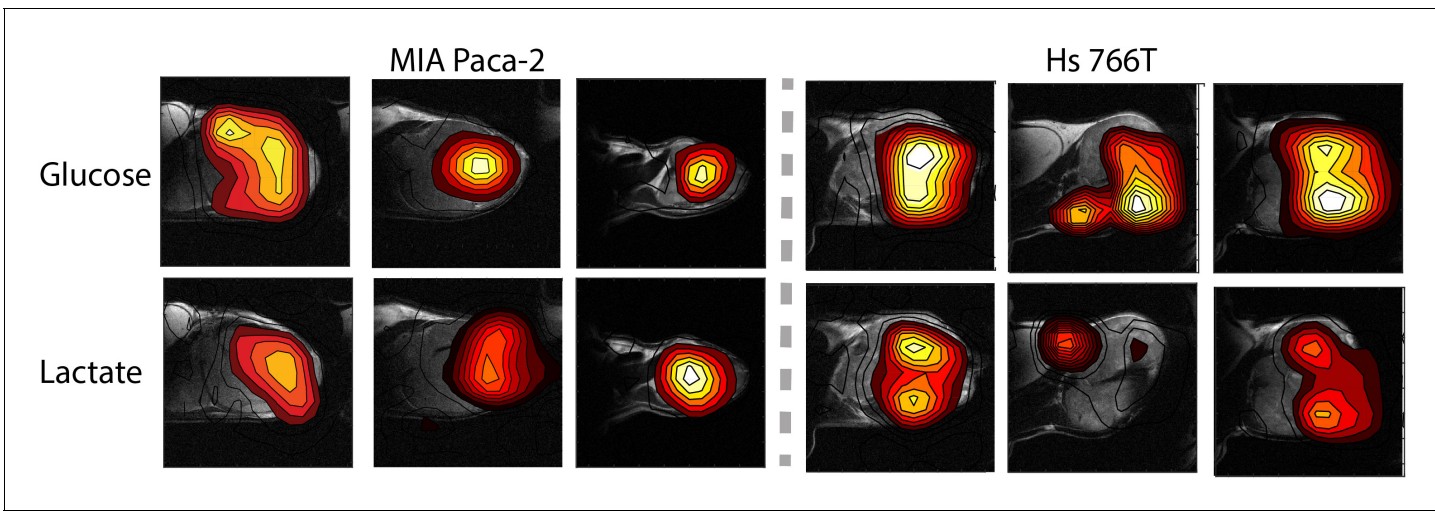

**Figure 5.** Contour maps created from time averages of the peak maxia of the glucose and lactate signals for three representative MIA Paca-2 (left) and Hs 766T (right) tumors.
DOI: https://doi.org/10.7554/eLife.46312.010

metabolism of tumors is not limited to the tumor itself, but also contains substantial contributions from the surrounding cells both directly through diffusion of metabolites across the tumor boundary (*Sousa et al., 2016*) and indirectly through the influence of regulatory and epigenetic signals (*Sherman et al., 2017*). The physical microenvironment of the tumor can also affect metabolism. Deficient or improperly formed (*Bailey et al., 2014*) vasculature often induces hypoxia in PDAC tumors (*Koong et al., 2000*) which can induce metabolic changes (*Matsumoto et al., 2014*; *Guillaumond et al., 2013*) that would not be evident by genetic analysis alone.

Methods for determining the internal metabolism of tumors non-invasively in vivo are lacking and represent a critical obstacle for the development of targeted metabolic therapy. The steady state metabolism of PDAC tumors can be probed indirectly through analysis of urine, blood, or pancreatic cyst fluid (*McGranaghan et al., 2016*) or more directly through magnetic resonance spectroscopy. However, metabolic networks are more flexible than protein networks and flux through the network can be rerouted to limit the impact of targeted enzymes (*Biancur et al., 2017*). Evaluating the target engagement of potential inhibitors can be difficult under these conditions. We demonstrate the potential of multimodal metabolic profiling of PDAC tumors for distinguishing xenograft tumors that are genetically similar but display very different metabolic phenotypes. Hs 766T is a cell line derived from lymphatic metastasis of pancreatic cancer that generates highly necrotic, hypoxic, slow growing heterogenous tumors. MIA Paca-2 is derived from primary cancer whose tumors are less necrotic, grow faster, and are highly homogenous. Despite their dissimilar origin and physiological differences, the steady state metabolism probed by CE/MS of Hs 766T and MIA Paca-2 is markedly similar, with only a few potential differentiating biomarkers (*Figure 1E*). Hyperpolarized pyruvate-lactate fluxes of Hs 766T and MIA Paca-2 xenografts estimated by $^{13}$C hyperpolarized pyruvate MRI were statistically indistinguishable (*Figure 3*). The flux through the TCA cycle and anaerobic fermentation is similar, as expected from the presence of *KRAS* and *TP53* mutations in both cell lines and similar LDHA levels (*Figure 1—figure supplement 1*). Hyperpolarized MRI of pyruvate, while it has proven clinically useful in other circumstances (*Gutte et al., 2015*), did not provide sufficient information for metabolic discrimination in this instance.

The use of [U-$^{13}$C] glucose instead of [1-$^{13}$C] pyruvate in this study allowed for a more comprehensive overview of metabolism. Measurements of in vivo glucose metabolism by $^{13}$C MRI have proved difficult because of difficulty of hyperpolarizing glucose and the low SNR in non-hyperpolarized experiments. The approximately 30 fold improvement in SNR from low rank tensor decomposition (*Figure 4E*) allows the difficult hyperpolarization step to be avoided (*Brender et al., 2019*), eliminating some of the practical barriers to clinical implementation of $^{13}$C metabolic imaging. The most important information acquired by this method is the direct measurement of rates of glycolysis and lactate fermentation, allowing imaging of the Warburg effect. High glucose uptake is one of the most well studied features of cancer and has been utilized in FDG-PET imaging in clinical settings. FDG-PET is limited in that the radiotracer cannot differentiate the metabolic conversions that occur beyond glucose uptake and phosphorylation. In contrast to FDG-PET, investigation of [U-$^{13}$C] glucose metabolism by $^{13}$C MRI can potentially probe more diverse and subtle metabolic differences in tumors. We observed one such metabolic difference in *Figure 3*; among Hs 766T and MIA Paca-2 PDAC xenograft animal models, glucose uptake was similar while glucose metabolism was distinct.

Our findings suggest some advantages for $^{13}$C glucose imaging. Compared to FDG-PET, there is no need for a radioactive tracer, which makes this imaging potentially safer and less invasive. By observing the lactate production at later time points, $^{13}$C glucose imaging can potentially detect cancer even in highly glucose-consuming tissue such as brain or liver. It can also potentially detect cancer in the bladder because glucose is not excreted immediately in urine, while FDG excreted in the urinary tract and excess signal in the bladder can interfere with lesion detection within or near the bladder wall. In contrast to hyperpolarized MRI, non-hyperpolarized $^{13}$C glucose imaging does not require onsite preparation of the probe, removing one of the main barriers to clinical translation of metabolic imaging by MRI.

Balancing these potential advantages, there are some limitations that should be acknowledged. The technique is inherently insensitive and requires the delivery of a large bolus that likely changes the initial metabolic state. The results therefore reflect the metabolism under a specific perturbation and should not be interpreted as reflective of the basal metabolic state. While this difference is likely less important in using metabolism as a biomarker, caution should be exercised in interpreting the results as reflective of the normal metabolism of the tissue. Other techniques, such as hyperpolarized

[13]C MRI, may be more useful in this context. The power of the low rank denoising technique scales as the square root of the number of acquisitions. As such, short repetition times are advantageous. The short repetition time skews the intensity towards metabolites with short T1 relaxation times. While the rates of conversion are not affected by this skew, the apparent concentrations are, and the relative intensities should not be used as a measure of relative concentration. While some challenges remain and the technique is inferior to PET in some respects, particularly with respect to resolution and imaging time, [13]C glucose imaging by MRI may emerge as a viable adjunct or alternative to FDG-PET in evaluation of patients with known or suspected cancer.

## Materials and methods

### Mouse models

The animal experiments were conducted according to a protocol approved by the Animal Research Advisory Committee of the NIH (RBB-159-2SA) in accordance with the National Institutes of Health Guidelines for Animal Research. Female athymic nude mice weighing approximately 26 g were supplied by the Frederick Cancer Research Center, Animal Production (Frederick, MD) and housed with ad libitum access to NIH Rodent Diet #31 Open Formula (Envigo) and water on a 12 hr light/dark cycle. Xenografts were generated by the subcutaneous injection of $3 \times 10^6$ MIA PaCa-2 (America Type Cell Collection (ATCC), Manassas, VA) or Hs 766T (Threshold Pharmaceuticals, Redwood City, CA) pancreatic ductal adenocarcinoma cells. (*Kim et al., 2009*) Both cell lines were tested in May 2013 and authenticated by IDEXX RADIL (Columbia, MO) using a panel of microsatellite markers. Molecular testing of cell lines for multiple pathogens, including mycoplasma, was performed at the time of receipt and prior to in vivo studies. All cell lines were maintained in RPMI 1640 supplemented with 10% fetal calf serum and antibiotics.

### CE/MS analysis

Tumors were excised when the volume reached 600 mm$^3$ and immediately frozen in liquid nitrogen and stored at $-80°C$ until analysis. A total of 116 metabolites involved in glycolysis, the pentose phosphate pathway, the tricarboxylic acid cycle, the urea cycle, and polyamine, creatine, purine, glutathione, nicotinamide, choline, and amino acid metabolism were analyzed using CE-TOF and QqQ mass spectrometry (Carcinoscope Package, Human Metabolome Technologies, Inc). Statistical significance was established by multiple two-sided t-tests, corrected for multiple comparisons by the two-stage linear step-up procedure of Holm et al with a confidence level of 5% (*Holm, 1979*; *Seaman et al., 1991*).

### Western blotting analysis

The mice bearing MIA Paca-2 and Hs 766T tumors (n = 4 for each group) were euthanized by breathing carbon dioxide gas, and tumor biopsy samples were excised. The tumor tissues were immediately homogenized with T-PER tissue protein extraction reagent (Thermo scientific). The homogenate was centrifuged, and the supernatant was used for western blot analysis. Hexokinase-2, Glut-1, LDHA proteins in tumor extract were separated on 4% to 20% Tris-Glycine gel and CD31 was separated on NuPAGE 3% to 8% Tris-Acetate gel (Life Technologies) by SDS-PAGE and were transferred to nitrocellulose membrane. The membranes were blocked for 1 hr in blocking buffer (3% nonfat dry milk in 0.1% Tween 20/TBS), which was then replaced by the primary antibody (1:500-1:1000), diluted in blocking buffer, and then incubated for 1 hr at room temperature. The membranes were then washed three times in washing buffer (0.1% Tween 20/TBS). The primary antibody was detected using horseradish peroxidase–linked goat anti-mouse or goat anti-rabbit IgG antibody at a 1:2000 dilution (Santa Cruz Biotechnology), visualized with Western Lightning Plus-ECL enhanced chemiluminescence substrate (Perkin Elmer Inc) and measured by the Fluor Chem HD2 chemiluminescent imaging system (Alpha Innotech Corp.). Density values for each protein were normalized to actin or HSC70.

### Hyperpolarized [13]C MRS experiments

Samples for NMR were prepared and analyzed as previously described in *Brender et al. (2019)* [1-[13]C] pyruvic acid (30 μL), containing 15 mM TAM and 2.5 mM gadolinium chelate ProHance

(Bracco Diagnostics, Milano, Italy), was hyperpolarized at 3.35 T and 1.4 K using the Hypersense DNP polarizer (Oxford Instruments, Abingdon, UK) according to the manufacturer's instructions. Typical polarization efficiencies were around 20%. After 40–60 min, the hyperpolarized sample was rapidly dissolved in 4.5 mL of a superheated HEPES based alkaline buffer. The dissolution buffer was neutralized with NaOH to pH 7.4. The hyperpolarized [1-$^{13}$C] pyruvate solution (96 mM) was intravenously injected through a catheter placed in the tail vein of the mouse (1.1 mmol/kg body weight). Hyperpolarized $^{13}$C MRI studies were performed on a 3 T scanner (MR Solutions, Guildford, UK) using a home-built $^{13}$C solenoid leg coil. After rapidly injecting the hyperpolarized [1-$^{13}$C] pyruvate, spectra were acquired every second for 240 s using a single pulse acquire sequence with a sweep width of 3.3 kHz and 256 FID points.

## Dynamic $^{13}$C glucose MRS without hyperpolarization

The magnetic resonance spectroscopy experiments were performed on a 9.4 T Biospec 94/30 horizontal scanner using a 16 mm double resonance $^{1}$H/$^{13}$C coil constructed as described in *Brender et al. (2019)*. Each mouse was anesthetized during imaging with isoflurane 1.5–2.0% administered as a gaseous mixture of 70% $N_2$ and 30% $O_2$ and kept warm using a circulating hot water bath. Both respiration and temperature were monitored continuously through the experiment and the degree of anesthesia adjusted to keep respiration and body temperature within a normal physiological range of 35–37° C and 60–90 breaths per min. Before the start of the experiment, anatomical images were acquired with a RARE fast spin echo sequence (*Hennig et al., 1986*) with 15 256 × 256 slices of 24 mm ×24 mm × 1 mm size with eight echoes per acquisition, a 3 s repetition time, and an effective sweep width of 50,000 Hz. Samples were shimmed to 20 Hz T with first and second order shims using the FASTMAP procedure (*Gruetter, 1993*).

[U-$^{13}$C] Glucose was administered as a 50 mg bolus injected into the tail vein immediately prior to the start of the experiment. Immediately after the injection, non-localized spectra were acquired with the NSPECT pulse-acquire sequence using maximum receiver gain, a repetition time of 50 ms, Ernst Angle excitation of 12°, 256 FID points, a sweep width of 198.6 ppm, 16 averages per scan, and 4500 scans for a total acquisition time of 1 hr. MLEV16 decoupling (*Levitt et al., 1983*; *Levitt et al., 1982*) was applied during acquisition using −20 dB of decoupling power and a 0.2 ms decoupling element. The decoupling pulse was centered on the main proton lipid resonance at 1.3 ppm. Chemical shift imaging experiments were performed similarly except an 8 × 8 image using 0.3 cm x 0.3 cm x 1.5 cm voxels was acquired every 48 s (16 averages per scan) for 90 min using rectilinear phase encoding.

## Signal processing

Signal processing was performed as described in *Brender et al. (2019)*. For non-localized (two dimensional) experiments, the first 67 points of the FID in the time dimension were removed to eliminate the distortion from the 13 ms dead time of the Bruker 9.4 T. (*Cobas, 2008*). The FID was Fourier transformed and the phase estimated by the entropy minimization method of *Chen et al. (2002)* as implemented in MatNMR (*van Beek, 2007*). After low rank reconstruction by SVD (see below), the baseline was estimated by a modification of the Dietrich first derivative method to generate a binary mask of baseline points (*Dietrich et al., 1991*), followed by spline interpolation using the Whittaker smoother (*Eilers, 2003*) to generate a smooth baseline curve (*Cobas et al., 2006*). The final correction adjusts for the limited number of points in the frequency dimension by continuation of the FID by linear prediction. The remaining 189 points of the FID after truncation in the first step were extrapolated to 1024 points using the 'forward-backward' linear prediction method of *Zhu and Bax (1992)*. Fourier transforming the FID of the transients from each voxel individually generated the final spectrum. Phase estimation proved difficult for to the chemical shift imaging experiments and therefore the spectra for chemical shift imaging experiments are shown in magnitude mode.

## Low rank reconstruction

For the two-dimensional signal matrices generated by non-localized pulse acquire experiments, the rank reduced signal was generated by truncating the SVD by setting the *N-r* diagonal values of the singular value matrix *S* to 0, where *N* is the number of rows in *S* and *r* is the predicted rank. The predicted rank was set to five unless otherwise specified. Tensor decomposition was achieved through

higher order orthogonal iteration (*De Lathauwer et al., 2000*) in the Matlab NWay package (*Andersson and Bro, 2000*) using a rank of 8 in the temporal and spectral dimensions and six in each spatial dimension.

## Kinetic modeling

Glucose kinetics were analyzed by piecewise analysis of the signal from glucose carbon-1 at 98 ppm (*Zierhut et al., 2010*). The signal at initial time points ($t_0 < 5$ m) can be described by a single, empirically based exponential transport function (*Chiang et al., 2017*) describing the combined effects of perfusion and import:

$$S_{98\,ppm}(t) = S_{98,total}\left(1 - Be^{-k_{trans}t}\right), \quad t_0 \tag{1}$$

Assuming negligible gluconeogenesis during the experiment, the kinetics after the complete passage of the bolus ($t > 5t_0 = 25$ m) can be approximated as a decay from multiple first order pathways:

$$\frac{dS_{98\,ppm}}{dt} = -\sum_i k_i, \qquad t > 5t_0 \tag{2}$$

which has the simple solution:

$$S_{98\,ppm}(t) = S_{98,total}\left(e^{-\left(\sum_i k_i\right)t}\right), \quad t > 5t_0 \tag{3}$$

where $k_i$ includes terms from both metabolic conversion and clearance of glucose from the tumor.

Since the system is not closed, the terms do not balance - the loss in glucose is not necessarily equal to the total change in the concentration of downstream metabolites because of clearance of glucose and other metabolites from the tumor. As such, it is difficult to evaluate the individual terms in *Equation 3* . We therefore report the sum $\sum_i k_i$ as the rate of glucose 'utilization', which we define as the time dependent change in glucose concentration in the field of view. Lactate kinetics are handled similarly except there is no transport term and the signal is evaluated at 23 ppm. Lactate 'formation' similarly refers to the time-dependent change in lactate concentration in the field of view and may reflect contributions from circulating [13]C-labeled lactate created outside the tumor (*Hui et al., 2017*; *Faubert et al., 2017*).

The kinetics of pyruvate metabolism in the hyperpolarized MRS experiment were evaluated by integrating the rate equations as described in *Daniels et al. (2016)*. For simplicity, a two-pool unidirectional flux model was assumed with equal relaxation times for lactate and pyruvate. The concentration of [13]C labeled lactate at the initial time point was also assumed to be zero. Making these assumptions, the final equations are:

$$Pyr(t) = \frac{Pyr(0)}{k}\left(e^{-\left(k+\frac{1}{T1}\right)t}\right), \quad t_0 \tag{4}$$

$$Lac(t) = \frac{Pyr(0)}{k}\left(e^{-\left(\frac{1}{T1}\right)t} - e^{-\left(k+\frac{1}{T1}\right)t}\right), \quad t_0 \tag{5}$$

## Ex vivo NMR analysis

50 mg of [13]C-labeled glucose was injected intravenously through the tail vein to start the labeling experiment. Mice were sacrificed by cervical dislocation one hour after injection, following an intravenous saline flush to reduce paramagnetic relaxation from heme contamination from the blood. The tumor was then removed and snap frozen in liquid nitrogen and then stored at −80°C until the extraction procedure.

The polar fractions were isolated from the frozen tumor sections using a modification of a previously published procedure for cell extracts (*Crooks et al., 2019*). Briefly, a section of the frozen tumor was cut and then pulverized in liquid nitrogen using a cryogenic grinder (Freezer/Mill 6875, Spex SamplePrep). Approximately 50 mg of the ground tissue powder was weighed and then immediately quenched with 2 ml of acetonitrile at −20°C. The solution was allowed to thaw on ice and 1.5

ml of ice cold dd $H_2O$ was added to the thawed extract. Lipids and non-polar metabolites were extracted by the addition of 1 ml of −20℃ chloroform with vigorous mixing. Addition of chloroform creates a three-phase system consisting of the polar and nonpolar fractions and an interphase layer consisting primarily of proteins. Following centrifugation at 6400 *g* for 30 min, 90% of the aqueous phase was transferred to a pre-tared microcentrifuge tube. After removal of the non-polar chloroform phase, the interphase layer was washed with ice cold 2:1 choloroform/methanol containing 1 mM BHT and recentrifuged. The aqueous phases were then combined and lyophilized. To remove residual proteins, the lyophilized powder was reconstituted in 100 µL of ice cold dd $H_2O$ followed by 400 µL of ice cold acetone. Samples were then incubated at −80℃ for 30 min to facilitate protein precipitation. The protein precipitate was isolated by centrifugation for 30 min at 14,000 rpm. The pellet was then washed with 100 µL of 60% acetonitrile and the remaining supernatant after centrifugation re-lyophilized.

NMR samples were prepared by dissolving the lyophilized powder in $D_2O$ containing DSS-d6 as reference and concentration standard. For each sample, a 1D $^1H$ Presat and 1D $^1H$-$^{13}C$ HSQC spectra were recorded at 15℃ on a 16.45 T Bruker Avance III spectrometer using a 1.7 mm inverse triple resonance cryoprobe with an acquisition time of 2 s and a relaxation delay of 4 s for the Presat experiment, and an acquisition time of 0.2 s and relaxation delay of 1.8 s for the HSQC experiments with adiabatic $^{13}C$ decoupling. The strongest sample from each group was also analyzed by high-resolution 2D multiplicity-edited HSQC and TOCSY (50 ms mixing time). TOCSY and high-resolution multiplicity-edited $^1H$-{$^{13}C$}-HSQC spectra were recorded at 14.1 T on an Agilent DD2 spectrometer using a 3 mm inverse triple resonance cold probe, with an acquisition time in $t_2$ of 1 s and a relaxation delay of 1 s and an isotropic mixing time of 50 ms (TOCSY) and an acquisition time of 0.2 s in $t_2$, recycle time of 2 s with 600 complex increments in $t_1$ for HSQC to resolve $^{13}C$-$^{13}C$ couplings in F1 (*Fan and Lane, 2016*).

Spectra were transformed, apodized using a cosine- squared function and 1 Hz line broadening exponential, phased and baseline corrected using the MNOVA software. Spectra were assigned using in-house data bases (*Fan and Lane, 2013*) and those of the HMDB (*Wishart et al., 2018*; *Wishart et al., 2007*) as previously described. 2D spectra were zero filled to 16384 × 2048 complex points (TOCSY), and 8192 × 4096 complex points (HSQC), transformed and apodized with a 1 Hz line broadening exponential and a cosine-squared function.

# Additional information

## Competing interests

Galen Reed, Albert P Chen, Jan Henrik Ardenkjaer-Larsen: is affiliated with GE HealthCare. The author has no other competing interests to declare. The other authors declare that no competing interests exist.

## Funding

| Funder | Grant reference number | Author |
| --- | --- | --- |
| National Cancer Institute | 1ZIASC006321-39 | James Mitchell |
| National Cancer Institute | Intramural Research Program | Murali C Krishna |
| Shared Resource(s) of the University of Kentucky Markey Cancer Center | P30CA177558 | Andrew N Lane Teresa WM Fan |

The funders had no role in study design, data collection and interpretation, or the decision to submit the work for publication.

## Author contributions

Shun Kishimoto, Jeffrey R Brender, Conceptualization, Formal analysis, Investigation, Methodology, Writing—original draft, Project administration, Writing—review and editing; Daniel R Crooks, Formal analysis, Investigation, Methodology, Project administration, Writing—review and editing; Shingo

Matsumoto, Conceptualization, Formal analysis, Investigation, Methodology, Writing—review and editing; Tomohiro Seki, Nobu Oshima, Penghui Lin, Jeeva Munasinghe, Keita Saito, Kazutoshi Yamamoto, Investigation; Hellmut Merkle, Resources, Methodology; Galen Reed, Albert P Chen, Jan Henrik Ardenkjaer-Larsen, Conceptualization, Writing—review and editing; Peter L Choyke, Funding acquisition, Writing—review and editing; James Mitchell, Resources, Funding acquisition; Andrew N Lane, Formal analysis, Supervision, Methodology, Writing—review and editing; Teresa WM Fan, Supervision, Funding acquisition, Writing—review and editing; W Marston Linehan, Funding acquisition; Murali C Krishna, Conceptualization, Funding acquisition, Writing—original draft, Writing—review and editing

### Author ORCIDs
Shun Kishimoto https://orcid.org/0000-0002-2496-5283
Jeffrey R Brender https://orcid.org/0000-0001-7487-6169
Murali C Krishna https://orcid.org/0000-0002-7216-7788

### Ethics
Animal experimentation: The animal experiments were conducted according to a protocol approved by the Animal Research Advisory Committee of the NIH (RBB-159-2SA) in accordance with the National Institutes of Health Guidelines for Animal Research.

### Decision letter and Author response
Decision letter https://doi.org/10.7554/eLife.46312.015
Author response https://doi.org/10.7554/eLife.46312.016

## Additional files

### Supplementary files
• Transparent reporting form
DOI: https://doi.org/10.7554/eLife.46312.011

### Data availability
Glucose imaging data and related files have been deposited to Dataverse at https://doi.org/10.7910/DVN/XU9XH9.

The following dataset was generated:

| Author(s) | Year | Dataset title | Dataset URL | Database and Identifier |
|-----------|------|---------------|-------------|-------------------------|
| Jeffrey R Brender | 2019 | 13C Glucose Imaging Raw Data | https://dataverse.harvard.edu/dataset.xhtml?persistentId=doi:10.7910/DVN/XU9XH9 | Harvard Dataverse, 10.7910/DVN/XU9XH9 |

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
