## [Decision Letter]

Thank you for submitting your article "Imaging of glucose metabolism by 13C-MRI distinguishes pancreatic cancer subtypes" for consideration by *eLife*. Your article has been reviewed by three peer reviewers, including Ralph DeBerardinis as the Reviewing Editor and Reviewer #1, and the evaluation has been overseen by a Reviewing Editor and Anna Akhmanova as the Senior Editor. The following individual involved in review of your submission has also agreed to reveal his identity: Kevin Brindle (Reviewer #2).

The reviewers have discussed the reviews with one another and the Reviewing Editor has drafted this decision to help you prepare a revised submission.

Summary:

Kishimoto et al., describe a technique to assess metabolism of non-hyperpolarized 13C-glucose metabolism in real time in pancreatic cancer xenografts. They inject [U-13C]glucose and detect glucose-derived metabolites kinetically using 13C MRS; applying rank reduction of noise improves detection of 13C-labeled metabolites and, when tensor decomposition is also used, allows some 13C features to be localized within the tumors. The authors use these techniques to a) characterize some of the metabolic fates of glucose within two PDAC xenograft models; and b) to identify differences between the models that could not be detected using hyperpolarized [1-13C]pyruvate imaging. A strength of the paper is the potential for this technique to estimate glucose-dependent metabolic fluxes non-invasively. Relative weaknesses in the current form include lack of benchmarking and inadequate description of some aspects of the technique.

Essential revisions:

1) More information about the experiments with [U-13C]glucose need to be included so that the reader can put the results into a physiological context. How was the glucose administered to the mice? A kinetic of glucose concentration and fractional enrichment should be provided to assess how the technique perturbs the basal metabolic state.

2) The metabolic differences between the two models depends on quantification of 3 resonances in vivo, a "predominantly glucose peak" at 95ppm, a "major glucose peak with contribution from all glycolytic intermediates" at 60ppm and the lactate resonance at 23ppm. Then ratios and rates for glucose and lactate are calculated. Details about how these calculations were made should be included. Wouldn't a rate for the 60ppm resonance be the most informative for glycolysis? Moreover, which metabolites contribute to the 60ppm resonance, and to what degree? Presumably with such a short acquisition time (50ms), differential relaxation would weight that resonance's magnitude to certain intermediates. Extracting metabolites after the experiment and examining their 13C enrichments may help clarify these resonances.

3) Along these lines, it would greatly increase confidence in the conclusions about metabolic differences between these tumors if the authors could use mass spectrometry or ex vivo spectroscopy to measure the total abundance and 13C enrichment in lactate from these tumors. This would allow them to benchmark their in vivo MRS data.

4) The authors state that the peak at 95 ppm can be assigned specifically to glucose and glucose-6-phosphate without contributions from other glycolytic intermediates. This needs a supporting reference or experimental evidence that this is the case. They go on to say that the intensity of the peak at 95 ppm therefore gives an indication of glucose import and phosphorylation. But the peak at 95 ppm surely includes contributions from intravascular and interstitial (i.e. extracellular) glucose, so it is unclear how this signal can be used to measure glucose uptake and phosphorylation.

5) From subsection “Glucose metabolism can be measured in vivo by 13C MRS: "The CE/MS data is suggestive of an upregulation in MiaPaCa-2 of the later stages of glycolysis relative to Hs766t, but the sample-sample variability inherent to MS techniques obscures the magnitude of any difference (see Figure 1—figure supplement 3). Hyperpolarized 13C MRS is more precise in this respect, but the transient nature of hyperpolarization restricts analysis to only the first few metabolic steps away from the probe". There are problems with these statements. First, it is not true that glycolysis cannot be measured using MS. Second, hyperpolarized 13C MRS is not more precise, and in fact it cannot measure concentrations because signal intensity is a function of concentration, T1 and time after injection of the hyperpolarized substrate. Third, it is also not true that hyperpolarization restricts analysis to only the first few metabolic steps away from the probe since conversion of glucose to lactate has been measured using in vivo hyperpolarization techniques.

6) The authors say that there is no difference in glucose uptake between the two xenografts or in the rate of lactate production. They then go onto say that Hs766t xenografts displayed slower glucose metabolism than MiaPaCa-2 xenografts. Given these data were obtained from a bolus injection of [1-13C]glucose, how do they reconcile this observation with no change in lactate production?

---

## [Author Response]

Essential revisions:1) More information about the experiments with [U-13C]glucose need to be included so that the reader can put the results into a physiological context. How was the glucose administered to the mice? A kinetic of glucose concentration and fractional enrichment should be provided to assess how the technique perturbs the basal metabolic state.

Glucose was administered as a 50 mg bolus injected into the tail vein immediately prior to the start of the experiment. We agree with reviewer that this large bolus raises blood glucose levels and perturbs the basal metabolic state. This is an inherent limitation of the technique. The last paragraph of the Discussion section has been modified to address these concerns and others.

“Balancing these potential advantages, there are some limitations that should be acknowledged. The technique is inherently insensitive and requires the delivery of a large bolus that likely changes the initial metabolic state. The results therefore reflect the metabolism under a specific perturbation and should not be interpreted as reflective of the basal metabolic state. While this difference is likely less important in using metabolism as a biomarker, caution should be exercised in interpreting the results as reflective of the normal metabolism of the tissue. Other techniques, such as hyperpolarized ^13^C MRI, may be more useful in this context. The power of the low rank denoising technique scales as the square root of the number of acquisitions. As such, short repetition times are advantageous. The short repetition time skews the intensity towards metabolites with short T1 relaxation rates. While the rates of conversion are not affected by this skew, the apparent concentrations are, and the relative intensities should not be used as a measure of relative concentration.”

*2) The metabolic differences between the two models depends on quantification of 3 resonances* in vivo*, a "predominantly glucose peak" at 95ppm, a "major glucose peak with contribution from all glycolytic intermediates" at 60ppm and the lactate resonance at 23ppm. Then ratios and rates for glucose and lactate are calculated.*

a) Details about how these calculations were made should be included.b) Wouldn't a rate for the 60ppm resonance be the most informative for glycolysis?c) Moreover, which metabolites contribute to the 60ppm resonance, and to what degree?d) Presumably with such a short acquisition time (50ms), differential relaxation would weight that resonance's magnitude to certain intermediates. Extracting metabolites after the experiment and examining their 13C enrichments may help clarify these resonances.

We agree with the reviewers that the details on why particular resonances were assigned and chosen for analysis are lacking in the original manuscript and appreciate the chance to clear up any possible confusion. In response to the specific queries:

a) A new section has been included in the Materials and methods section to include the details of the analysis.

*“*Glucose kinetics were analyzed by piecewise analysis of the signal from glucose carbon-1 at 98 ppm.*^70^* The signal at initial time points (t_0_ <5 m) can be described by a single, empirically based exponential transport function*^71^* describing the combined effects of perfusion and import:

S98ppmt=S98,total1-Be-ktranst,t<t0 Eq. 1

Assuming negligible gluconeogenesis during the experiment, the kinetics after the complete passage of the bolus (t > 5t_0_ = 25 m) can be approximated as a decay from multiple first order pathways:

dS98ppmdt=-∑iki,t>5t0 Eq. 2

which then has the simple solution:

S98ppmt=S98,totale-∑ikit,t>5t0 Eq. 3

where k_i_ includes terms from both metabolic conversion and clearance of glucose from the tumor.

Since the system is not closed, the terms do not balance -the loss in glucose is not necessarily equal to the total change in the concentration of downstream metabolites because of clearance of glucose from the tumor. As such, it is difficult to evaluate the individual terms in Eq. 3. We therefore report the sum ∑iki as the rate of glucose “utilization”, which we define as the time dependent change in glucose concentration in the field of view. Lactate kinetics are handled similarly except there is no transport term and the signal is evaluated at 23 ppm. Lactate “formation” similarly refers to the time dependent change in lactate concentration in the field of view and may reflect contributions from circulating ^13^C-labeled lactate created outside the tumor.

The kinetics of pyruvate metabolism in the hyperpolarized MRS experiment is evaluated by integrating the rate equations as described in Ref. *^72^*. For simplicity, a two-pool unidirectional flux model was assumed with equal relaxation times for lactate and pyruvate. The concentration of ^13^C labelled lactate at the initial time point was also assumed to be zero. Making these assumptions, the final equations are:

Pyrt=Pyr0ke-k+1T1t,t>t0 Eq. 4

Lact=Pyr0ke-1T1t-e-k+1T1t,t>t0 Eq. 5”

b and c) The peak at 60 ppm resonance potentially contains resonances from several glycolytic intermediates. However, not all glycolytic intermediates have peaks in this range, which considerably complicates analysis. The peaks at 95 and 98 ppm only arise from glucose and glucose-6-phosphate and, as described below, glucose-6-phosphate was not found in detectable concentrations. This considerably facilitates analysis and the peak at 98 ppm was used consistently for all analysis.

To confirm assignment, we carried out a series of ex vivo NMR experiments as described in the revised paragraph below:

**“**Using this technique, we first checked the glucose metabolism of each tumor type following an injection of 50 mg bolus of [U-^13^C] glucose using non-localized spectroscopy. The resulting spectra are complex and include contributions from the α and β anomers of glucose, the natural abundance ^13^C signal from lipids, as well as signals from downstream metabolites such as lactate and alanine (Figure 3A). To help resolve these ambiguities, tumors were flash frozen 1 hour after the injection of a [U-^13^C] glucose bolus and the polar fraction analyzed by multidimensional NMR. Strong signals at 22.8 and 18.9 ppm in the indirectdimension of the HSQC spectrum (Figure 3—figure supplement 1A) and carbon satellites in the TOCSY spectrum*^30^* (Figure 3—figure supplement 1B) confirmed that the peaks at 22.8 ppm and 18.9 ppm arise from ^13^C labeled lactate and alanine, respectively (Figure 1—figure supplement 3). ^13^C labeled glutamate and aspartate are detectable in the ex vivo HSQC but not in the in vivo MRS, likely because the longer relaxation times*^31, 32^* strongly attenuates the signal with the short recycle delay used in the in vivo experiment. The peaks at 95 and 98 ppm can potentially arise from carbon 1 of the α and β anomers of either glucose or glucose-6-phosphate. The absence of any detectable peaks near 75 ppm confirms they arise exclusively from glucose without contribution from glucose-6-phosphate any other glycolytic intermediates. While the peaks at 95 and 98 ppm can be definitively assigned to carbon 1 of glucose, the HSQC and the pre-injection spectra (Figure 3A) confirms the other intense peak at 63 ppm and the spectrally crowded region between 63 and 78 ppm contains contributions from glycerol containing species. The lack of ^13^C-^13^C couplings for the glycerol resonances in the HSQC suggest these molecules are not derived directly from the glucose bolus.

The peak at 98 ppm was therefore used as a marker as it can be assigned specifically in this case to glucose and not any other molecule. Specifically, an increase in the intensity of the 98 ppm reflects the arrival of glucose from the bloodstream into the instrument’s field of view (gluconeogenesis within the tumor is assumed to be negligible at this time scale). A decrease in intensity reflects either the removal of glucose by the bloodstream out of the field of view or the conversion of glucose into another species. Similarly, the appearance of peaks at 22.8 and 18.9 ppm reflect the conversion of lactate and alanine, respectively, or the arrival of circulating lactate or alanine produced from ^13^C glucose outside the tumor.*^33, 34^*”

d) The reviewers are correct that shorter relaxation times can bias the peak intensity towards certain intermediates. The relaxation time, however, does not affect the corresponding rate. The experiment is done at thermal equilibrium with respective to the polarization and the signal does not evolve in the absence of perfusion or metabolism. Since the overall relaxation is constant throughout the experiment, it has no effect on the calculated rate, which is the quantity being measured.

*3) Along these lines, it would greatly increase confidence in the conclusions about metabolic differences between these tumors if the authors could use mass spectrometry or* ex vivo *spectroscopy to measure the total abundance and 13C enrichment in lactate from these tumors. This would allow them to benchmark their* in vivo *MRS data.*

We agree with the reviewers that benchmarking to an established method is an important part of the validation of any new technique. Toward this end, we spent considerable effort in attempting to quantify the ^13^C enrichment of glucose and lactate in the polar fraction of the tumor extracts. Progress was complicated by the low concentration of glucose in the intracellular and interstitial fluid, which hindered observation of ^13^C satellites of glucose in the TOCSY and ^1^H spectra and complicated measurement of absolute concentrations or fractional enrichments. While establishing absolute concentrations was difficult, we could successfully measure the ratio of key metabolites from the HSQC spectra. As expected, the metabolite ratios from ex vivo NMR do not match those observed in the in vivo MRS technique because of the strong bias towards fast relaxing species such as glucose. Nevertheless, the predicted trend (higher lactate to glucose ratios in MiaPaca) is similar in both experiments. Glucose concentrations were also much more variable in MiaPaca, in agreement with the higher variance observed in Figure 3E.

*“*This difference is also reflected in the time-averaged glucose to lactate ratio (Figure 3F, Mann-Whitney rank test, p=0.03, Cohen’s d = 1.20, large effect size), which is an approximate measure of the relative rates of the appearance of lactate and disappearance of glucose within the FOV of the scanner.*^36^* To confirm the presence of a metabolic difference, the glucose, lactate, and glutamate peaks of the HSQC spectra from the polar extracts of MIA Paca-2 and Hs 766T tumors were quantified (Figure 3H). As expected, the metabolite ratios from ex vivo NMR do not match those observed in the in vivo MRS technique because of the strong bias towards fast relaxing species such as glucose in the MRS spectrum due to the short repetition time. Nevertheless, the predicted trend (higher lactate to glucose ratios in MIA Paca-2) is similar in both experiments.”

MIA Paca-2Hs766tSpeciesRatio meansemRatio meansem^13^C Lac-3/^13^C Ala-34.480.262.92014^13^C Glu 4/^13^CLac-30.160.010.640.085^13^C Lac-3/^13^Glc-15.640.983.760.43

Metabolite ratios obtained by peak integration of resonances in the HSQC spectra from obtained from polar extracts of the tumor xenografts. A 50 mg bolus of [U-13C] glucose was injected intravenously and the peaks corresponding to each resonance in the HSQC spectrum were integrated to obtain the ratio. Five mice were used for each sample. Unpaired t test on means: Lac/Ala p=0000706 (t=5.33), Glu4/Lac p=0.000485 (t=-5.643), and Lac/Glc p=0.086 (t=2.05).”

4) The authors state that the peak at 95 ppm can be assigned specifically to glucose and glucose-6-phosphate without contributions from other glycolytic intermediates. This needs a supporting reference or experimental evidence that this is the case. They go on to say that the intensity of the peak at 95 ppm therefore gives an indication of glucose import and phosphorylation. But the peak at 95 ppm surely includes contributions from intravascular and interstitial (i.e. extracellular) glucose, so it is unclear how this signal can be used to measure glucose uptake and phosphorylation.

We have confirmed the assignment using ex vivo NMR as described in the response to comment 3.The reference to 95 ppm containing potential contributions from the β-anomers glucose and glucose comes from analysis of the Human Metabolome Database. However, the actual concentration of glucose-6-phosphate was found to be negligible, in line with the steady state results from CE/MS.

The reviewers are correct in pointing out that the observed signal has contributions from everything within the FOV. The intensity of the peaks at 95 and 98 ppm are a measure of the relative concentration of glucose and glucose-6-phosphate within the FOV at a given time point. Since within this sample, the concentration of glucose-6-phosphate was found to negligible, the intensity of the 95/98 ppm peaks is a measure of the relative glucose concentration specifically. As described in our response to Comment 3, anything that changes the glucose concentration within the FOV, including a change in intravascular and interstitial glucose concentrations due to perfusion, will affect the signal.

*5) From subsection “Glucose metabolism can be measured* in vivo *by 13C MRS: "The CE/MS data is suggestive of an upregulation in MiaPaCa-2 of the later stages of glycolysis relative to Hs766t, but the sample-sample variability inherent to MS techniques obscures the magnitude of any difference (see Figure 1—figure supplement 3). Hyperpolarized 13C MRS is more precise in this respect, but the transient nature of hyperpolarization restricts analysis to only the first few metabolic steps away from the probe". There are problems with these statements.*

a) First, it is not true that glycolysis cannot be measured using MS.b) Second, hyperpolarized 13C MRS is not more precise, and in fact it cannot measure concentrations because signal intensity is a function of concentration, T1 and time after injection of the hyperpolarized substrate.

*c) Third, it is also not true that hyperpolarization restricts analysis to only the first few metabolic steps away from the probe since conversion of glucose to lactate has been measured using* in vivo *hyperpolarization techniques.*

We agree with reviewers that this statement is not precise and contains several erroneous and problematic statements. Accordingly, we have rewrote this section to more clearly convey our intent:

“The CE/MS data is suggestive of an upregulation in MIA Paca-2 of the later stages of glycolysis relative to Hs 766T (Figure 1—figure supplement 3), although no single metabolite in the glycolytic pathway stands out as being statistically different (Figure 1). The CE/MS assay, however, requires a biopsy, which may be undesirable in some circumstances. To probe the glycolytic pathway in vivo, a different technique is needed.

Hyperpolarized glucose imaging has been successfully used to image glycolysis in vivo. While glucose is an exemplary tracer from a biological standpoint and has been shown to be very useful in pre-clinical hyperpolarized MRI research,*^1^* the extent of hyperpolarization achieved places constraints that may be difficult to realize in some situations in a clinical setting. Hyperpolarization of ^13^C tracers has been considered necessary for imaging and kinetic studies due to the inherently low signal of ^13^C MRS. If noise can be reduced to acceptable levels without hyperpolarization, the restrictions hyperpolarization places on an experiment can be removed. "

6) The authors say that there is no difference in glucose uptake between the two xenografts or in the rate of lactate production. They then go onto say that Hs766t xenografts displayed slower glucose metabolism than MiaPaCa-2 xenografts. Given these data were obtained from a bolus injection of [1-13C]glucose, how do they reconcile this observation with no change in lactate production?

We thank the reviewers for bringing out this important point. There are multiple possible pathways for glucose to be metabolized. The end product for glucose metabolized through the OXPHOS pathway is CO_2_, which would not be detectable by our technique which is skewed towards detection of species with short relaxation times. Also, circulating ^13^C lactate may enter the tumor through external sources, which would remove the apparent requirement for balance.